# Structural Determinants of Substrate Specificity of Omega-3 Desaturases from *Mortierella alpina* and *Rhizophagus irregularis* by Domain-Swapping and Molecular Docking

**DOI:** 10.3390/ijms20071603

**Published:** 2019-03-30

**Authors:** Chunchi Rong, Haiqin Chen, Xin Tang, Zhennan Gu, Jianxin Zhao, Hao Zhang, Yongquan Chen, Wei Chen

**Affiliations:** 1State Key Laboratory of Food Science and Technology, Jiangnan University, Wuxi 214122, China; 7130112043@vip.jiangnan.edu.cn (C.R.); xintang@jiangnan.edu.cn (X.T.); zhennangu@jiangnan.edu.cn (Z.G.); jxzhao@jiangnan.edu.cn (J.Z.); zhanghao@jiangnan.edu.cn (H.Z.); yqchen@jiangnan.edu.cn (Y.C.); weichen@jiangnan.edu.cn (W.C.); 2School of Food Science and Technology, Jiangnan University, Wuxi 214122, China; 3Department of Cancer Biology, Wake Forest School of Medicine, Winston-Salem, NC 5: 27127, USA; 4National Engineering Research Center for Functional Food, Jiangnan University, Wuxi 214122, China; 5Beijing Innovation Center of Food Nutrition and Human Health, Beijing Technology and Business University (BTBU), Beijing 100048, China

**Keywords:** linoleic acid, arachidonic acid, omega-3 desaturases, chimera, domain-swapping, molecular docking

## Abstract

Although various ω-3 fatty acid desaturases (ω3Des) have been identified and well-studied regarding substrate preference and regiospecificity, the molecular mechanism of their substrate specificities remains to be investigated. Here we compared two ω3Des, FADS15 from *Mortierella alpina* and oRiFADS17 from *Rhizophagus irregularis*, which possessed a substrate preference for linoleic acid and arachidonic acid, respectively. Their sequences were divided into six sections and a domain-swapping strategy was used to test the role of each section in catalytic activity. Heterologous expression and fatty acid experiments of hybrid enzymes in *Saccharomyces cerevisiae INVSc1* indicated that the sequences between his-boxes I and II played critical roles in influencing substrate preference. Based on site-directed mutagenesis and molecular docking, the amino acid substitutions W129T and T144W, located in the upper part of the hydrocarbon chain, were found to be involved in substrate specificity, while V137T and V152T were confirmed to interfere with substrate recognition. This study provides significant insight into the structure-function relationship of ω3Des.

## 1. Introduction

ω-3 Fatty acid desaturases (ω3Des) are key enzymes in the synthesis of polyunsaturated fatty acids (PUFAs) in the ω-6 and ω-3 pathways [1]. These enzymes convert ω-6 PUFAs to their ω-3 counterparts by removing two hydrogen atoms and adding a double bond in the fatty acid hydrocarbon chain and possess considerable nutritional and medical value by their ability to control the composition of storage lipids. They have been used to produce eicosapentaenoic acid (EPA, C20:5^Δ5,8,11,14,17^) and docosahexaenoic acid (DHA, C22:6^Δ4,7,10,13,16,19^) in oleaginous microorganisms, which are important for brain development, tissue formation and repair in mammals and also have significant effects in the prevention of asthma, cancer, depression, obesity, and immune cardiovascular disorder [2,3,4].

Various ω3Des from a wide range of species have been found to exhibit obvious substrate specificity with divergence in catalytic capability and preference towards the presence of various fatty acid substrates [5,6]. A Δ15 substrate preference for the linoleic acid (LA, C18:2^Δ9,12^) and γ-linolenic acid (GLA, C18:3^Δ6,9,12^) substrates was observed in ω3Des, like the yeast desaturase from *Saccharomyces kluyveri* [7,8]; bifunctional Δ12/ω3 desaturases from *Fusarium moniliforme*, *Fusarium graminearum*, and *Magnaporthe grisea* [9]; FADS15 [10] and maw3 [11] from *Mortierella alpina*, ShFAD3 from *Salvia hispanica*, and PfFAD3 from *Perilla frutescens* [12]. In contrast, a strong Δ17 desaturase preference towards the arachidonic acid (AA, C20:4^Δ5,8,11,14^) substrate was found within another series of ω3Des, such as sdd17 from *Saprolegnia diclina* [13], Pi-D17 from *Phytophthora infestans* [14], oPPD17 from *Phytophthora parasitica* [15], and three bifunctional Δ17Des from *Pythium aphanidermatum, Phytophthora sojae and Phytophthora ramorum* [16].

Various Δ4, Δ5, Δ6, Δ8, Δ9, and Δ12 fatty acid desaturases have also been identified, and their substrate preferences and regiospecificities have been well-studied. By systematic analysis, a membrane-peripheral region close to the catalytic center was revealed to cause functional exchange between Δ12Des and Δ12/ω3Des from *Aspergillus nidulans* in 2007 [17]. A considerable functional divergence in substrate specificity was clearly demonstrated in two Δ6Des isolated from *Mortierella alpina* and *Micromonas pusilla* [18], respectively. For the *Siganus canaliculatus* Δ4 and Δ5/6 desaturases, the heterologous expression of the corresponding constructed chimeras and study of mutation effects indicated that single or double amino acid substitutions were sufficient to alter the desaturation activities [19], and the same was also found in *Borago officinalis* Δ6Des and Δ8Des [20], and the house cricket Δ12/Δ9Des [21]. Furthermore, the tyrosine within the pocket of Δ9Des (ChDes9-1 and ChDes9-2) was found to strongly control the chain length specificity of the substrates in the presence of highly specific enzymes from the marine copepod *Calanus hyperboreus* [22]. The fundamental mechanistic characteristics of all of these novel desaturases have been elucidated through systematic fragment exchange. However, the structure-function relationships of membrane-bound ω3Des have not been well understood. 

In this study, we compared two ω3Des, FADS15 (GenBank accession number KF433065) from an AA-enriched strain of *Mortierella alpina* and oRiFADS17 (GenBank accession number MH028784) from a fatty acid auxotroph of *Rhizophagus irregularis*. They possessed a high desaturation activity towards LA and AA substrates, respectively. However, they were very different in substrate specificity. To evaluate the precise region that determined the substrate specificity of these two enzymes, a set of chimeric enzymes were constructed in a *Saccharomyces cerevisiae* (*S. cerevisiae*) *INVSc1* expression system using a domain-swapping strategy. Then, eight mutants of the two ω3Des localized at the crucial region were generated, guided by multiple sequence alignment. Finally, based on molecular docking analysis, the crucial amino acids and their roles in regulating substrate specificity and enzyme activity were identified. This study visualized the proposed structure of ω3Des for the first time and provided significant insight into the relationship between the structure and function of ω3Des.

## 2. Results

### 2.1. Similar Sequences and Topological Structures of FADS15 and oRiFADS17

The biosynthetic process of long-chain PUFAs can be classified into the alternative Δ8-pathway (Δ9Elo/Δ8Des/ω3Des) and the conventional Δ6-pathway (Δ6Des/Δ6Elo/ω3Des), as illustrated in Appendix A. The Δ8-pathway is an alternative pathway, which has been identified in fungi, plants, animals, etc., and the Δ6-pathway is the predominant pathway mostly found in algae, mosses, fungi, nematodes, fish, and animals [5,6]. ω3Des are key enzymes for both pathways and crucial for EPA and DHA production in combination with a series of other desaturases and elongases.

FADS15 and oRiFADS17 are typical ω3Des, in that they tolerate a great diversity of substrates. Sequence analysis of the two full cDNAs indicated that FADS15 and oRiFADS17 encoded polypeptides of 403 and 397 amino acids, respectively, which shared at least 49% amino acid sequence similarity. To further elucidate their membrane structures, topology analysis of these two ω3Des was carried out using the Clustal Omega Web servers. As shown in Figure 1 and Figure 2A, the structural assemblies of their amino acid sequences were highly similar and had three conserved regions of his-box motifs, which were predicted to be di-iron binding sites situated at the center of the substrate-binding pocket. The highly conserved amino acids, which were indicated by light shading, may be involved in substrate recognition (Figure 1). Their membrane structures contained four large hydrophobic domains, two membrane-embedded helices and seven separate hydrophilic clusters with the N- and C-termini facing the cytosol, like in most novel desaturases (Figure 2A).

### 2.2. Identification of Determinants of Substrate Specificity in 12 Chimeras

To confirm the functions of FADS15 and oRiFADS17, their coding regions were individually subcloned into a yeast expression vector pYES2/NT C, and the resulting plasmids were then introduced into the *S. cerevisiae INVSc1* strain. The fatty acid experiments indicated that the two desaturases had a broad ω-6 fatty acid substrate spectrum and were capable of converting the supplemented ω-6 PUFA substrates, including LA, GLA, DGLA, and AA (Table 1). Furthermore, it was shown that FADS15 possessed a higher catalytic activity towards LA and GLA, whereas oRiFADS17 possessed a higher catalytic activity towards DGLA and AA. In addition, FADS15 favored LA over AA with an activity ratio of 2.24 and oRiFADS17 had an activity ratio of 0.631, thus exhibiting a strong Δ15 and Δ17 desaturase preference, respectively.

To investigate the determinants that regulated substrate preference or enzyme activity, we compared the membrane structures of the two ω3Des. Domains of chimeric enzymes were designed and systematically swapped by their highly homologous primary structures (Figure 2A and Figure 3). The fatty acid compositions of the fusion enzymes were analyzed and their individual desaturation activities were estimated as the ratio of fatty acids in the substrates and products (Table 2). The resulting conversion rates, given in Figure 4B, showed that C1 and C7 could catalyze LA and AA, but their conversion rates were much lower than those of FADS15 and oRiFADS17. A similar finding was obtained in both C6 and C12. For C12, no ALA was generated at a detectable level when LA was added to the medium, and the conversion rate of AA was also significantly lower. These results suggested that the N- and C-terminal areas of both ω3Des (hydrophilic regions A and F) had an important role in desaturase activity, but did not participate in specifying the substrate chain length.

To examine the role in the substrate specificity played by the hydrophobic domains (regions B and E) of the two parent proteins, the hybrid enzymes C2, C5, C8, and C11 were designed and generated using a mixture of primers (Appendix A). In GC-MS analysis of the fatty acids in the resultant chimeras, neither ALA nor EPA was detected in all cases. It can be reasonably supposed that the mutual replacement of the large hydrophobic domains contributes to conformational instability, thus resulting in null desaturase activities of the ω3Des.

In addition, membrane-embedded helices C and D (hydrophobic region along with peripheral linkers), which were separated by the second his-box motif, were exchanged in the corresponding chimeras to identify the crucial region that influenced substrate specificity. As illustrated in Figure 4B, the results showed that substrate specificity of C3 was remarkably reduced and its conversion level dropped by approximately 45% towards LA but 8% towards AA. The substrate preference of the reverse enzyme C9 was also altered when 36 of amino acids of oRiFADS17 were replaced with the analogous amino acids of FADS15. Collectively, these results indicated that the third hydrophobic region (domain C) critically affected the substrate specificity of the ω3Des, thus resulting in the preference towards substrates of different chain lengths. Meanwhile, the conversion rates of the C10 constructs towards both LA and AA dropped to approximately half that of the wild-type desaturase while the same substrate specificity was maintained. The C4 enzyme had no desaturase activity towards AA and its conversion rate of LA was greatly reduced to 13%, suggesting that domain D may play an important role in substrate recognition but does not regulate substrate specificity.

### 2.3. Mutation Study of Amino Acids Responsible for Substrate Specificity

To further discern the function of hydrophobic region C (*M. alpina*: amino acid residues 115–150; *R. irregularis*: amino acid residues 130–165) and determine which amino acid or acids were critical, specific residues were chosen, guided by multiple sequence alignment, and switched reciprocally between the two ω3Des as targets for functional modification (Figure 1 and Figure 5A). Next, a series of distinct mutants were generated by site-directed mutagenesis. Expression analysis of their yeast transformants showed that the conversion rates of mutants Y139F and S145T in FADS15 and Y154F and S160T in oRiFADS17 retained the same desaturase activity towards both LA and AA substrates, implying that these sites were not required for substrate specificity (Table 3). However, the catalytic activity was significantly reduced by the substitution of valine by threonine (V137T and V152T) at the putative substrate-binding pocket. Furthermore, the valine positioned at 152 appeared to be the more important of the two for the desaturation activity of oRiFADS17 according to their conversion rates in Figure 5B. In contrast, the substrate selectivity of the W129T and T144W mutants was greatly changed compared with the wild-type, and the conversion rates of W129T towards AA increased by nearly 25%. The results indicated that these tryptophan/threonine residues at the corresponding positions had a profound impact on the substrate discrimination of these two enzymes.

### 2.4. Homology Modelling and Molecular Docking Analysis of FADS15 and oRiFADS17

To interpret the fatty acid results of the mutants, homology modeling and molecular docking of FADS15 and oRiFADS17 were carried out by the recent crystal structure of stearoyl-CoA desaturase hSCD1 as a template (Figure 6A–E). The representative substrates LA-CoA and AA-CoA were chosen as the ligands (Figure 2B) and captures of the tunnel-like pockets were shown in the hydrophobic interior of the ω3Des (Figure 6C,F). The mutations V137T and V152T adjacent to the active center were presumed to interfere with the interactions in the binding sites between substrates and protein (Figure 6G,H). Meanwhile, the crucial mutations W129T and T144W, located in the upper part of the hydrocarbon chain were closer to the predicted zinc-ion active center and were presumed to contribute to substrate specificity by affecting substrate localization (Figure 6G,F). The other four mutated residues, downstream of W129 of FADS15 and T144 of oRiFADS17, were also near the binding pockets, but probably did not face the substrate side.

## 3. Discussion

Although various n-3 fatty acid desaturases have been identified, few Δ17Des have been isolated compared to reported Δ15Des. The reported Δ17Des such as sdd17 [13], Pi-D17 [14], and oPPD17 [15] show moderate similarities in their primary sequences with FADS15 (Δ15Des). However, they still could not be applied to conduct domain-swapping with FADS15 because of their differences in lengths of amino acid sequences. Despite considerable efforts to identify and characterize ω3Des from various species (Appendix A), the underlying cause of the distinct substrate specificity of ω3Des has not been elucidated due to difficulties in separation and purification of large quantities of membrane protein and a lack of 3-dimensional crystal structure information of ω3Des.

The *M. alpina* FADS15 (Δ15Des) shares at least 49% amino acid identity with the *R. irregularis* oRiFADS17 (Δ17Des) and the two show high similarities in their primary sequences. In this research, they were confirmed to possess high fatty acid conversion rates and have a preference for C18 and C20 substrates, respectively (Figure 4A), when heterologously expressed in *S. cerevisiae*. To identify the factors responsible for the substrate discrimination of the two enzymes, their amino acid sequences were systematically swapped (Figure 3), and a set of fusion fragments were generated based on secondary structure predictions.

Previous researchers have reported that the C-terminal amino acid residues of Δ6Des and Δ8Des are involved in substrate recognition or catalysis activity [23,24] and that the N-terminal region of Δ5Des is related to desaturation activity [20,25]. In this study, we used the two representative LA and AA substrates to analyze the desaturase function of the N- or C-terminal domains based on the fused enzymes C1, C6, C7, and C12 (Figure 4B). The results indicated that these terminal areas of both enzymes did not affect substrate specificity, but may act on their maximum activity.

The transmembrane helix in hydrophobic region B of two marine copepod Δ9Des [22] and that in region E of the *S. canaliculatus* Δ4 and Δ5/6 desaturases [17] have been proposed to influence the chain length of the substrates. By comparing the activities of the fusion proteins C2, C5, C8, and C11, the substitution of these large hydrophobic domains was found to result in a loss of desaturase activity in all cases (Figure 4B). It is reasonable to assume that the null activity was due to the disruption of the overall structure of the two enzymes. This finding suggested that domains B and E played an important role in maintaining the conformational stability of the enzyme structure, and that replacing the main structural domains of these enzymes with the corresponding domains in the other enzyme, by analogy with previous studies [17,18,19,20,21,22], may disrupt the overall structure of these chimeric enzymes.

The regions of domains C and E near the active center were assumed to be responsible for determining the substrate regioselectivity of Δ12Des and bifunctional Δ12/ω3Des from *A. nidulans* [17]. To test the effects of the putative regions, chimeric enzymes C3, C4, C9, and C10 were constructed and heterologously expressed (Figure 4B). The results demonstrated that both domains were involved in substrate binding and catalysis activity, and the third hydrophobic region significantly altered their substrate preferences when the intrinsic sequences were mutually replaced.

Based on molecular docking analysis, it was shown that the substitution of amino acid residues V137 and V152 might interfere with substrate recognition and desaturation activity (Figure 5B). The substitution of these valines was presumed to weaken the hydrophobic affinity with the acyl chain of the acetyl-CoA substrate, while the hydroxyl group of threonine allowed interaction with other polar groups, thus restricting insertion of the acyl chain. The exchange of tryptophan and threonine located in the upper part of the hydrocarbon chain was confirmed to affect substrate specificity (Figure 6G,H). It was speculated that the W129T mutation of FADS15 created more space for the substrate to enter into or be released from the tunnel and allowed the substrate acyl chain to be inserted much deeper when the large and hydrophobic tryptophan was switched to threonine. The other four nonsense mutations downstream of 129 probably did not face the substrate side, like these invalid amino acid residues described in the marine copepod Δ9Des [22].

In a previous paper, we used sequence comparisons to evaluate eight residues that might contribute to substrate preference of FADS15 [10]. Three valuable residues of these sites were found, which changed the LA/AA activity ratio, achieving almost as much as the domain C. However, these mutations would be nonsense when mutated together with other residues in their corresponding domains, and the simulation model was not successfully docked with the substrate ligand. In this study, some new research results were obtained. Based on a domain-swapping strategy, it was first revealed that the region C caused relatively modest changes in substrate preference (The ration of LA/AA activities shifted from 0.631 to 0.907 for oRiFADS17, and the change for FADS15 was from 2.24 to 1.303). During the fatty acid experiment, the reciprocal substitutions W129T in FADS15 and reciprocal T144W in oRiFADS17 in region C shifted the substrate preferences of the enzymes to 1.63 for FADS15 and to 0.854 for oRiFADS17, respectively. Depending on successful molecular docking, it was also first visualized that how the proposed structure of ω3Des interacted with fatty acid substrates and would contribute to evaluating the crucial roles of active amino acids in regulating substrate specificity and enzyme activity of desaturases or other important enzymes.

The traditional source of n-3 PUFAs is diminishing due to overfishing and marine pollution, and a low dietary intake of n-3 PUFAs, especially EPA and DHA, is observed worldwide. The knowledge of the relationship between the structure and function of ω3Des could be conducive for the molecular modification of desaturases and genetic engineering of oleaginous microorganisms, which therefore contributes to producing PUFAs for dietary supplementation or the prevention and treatment of numerous diseases.

## 4. Materials and Methods

### 4.1. Strains and Plasmids

The plasmids pYES2-FADS15 and pYES2-oRiFADS17 were stored in *Escherichia coli* Top 10 in our laboratory [10]. The *S. cerevisiae INVSc1* yeast strain was obtained from Invitrogen (Shanghai, China). The plasmid pYES2/NT C (Invitrogen, Shanghai, China) was used for heterologous expression of the parent proteins, 12 chimeras, and their mutants.

### 4.2. Materials

*Escherichia coli* Top 10 containing the plasmids of the parent proteins, 12 recombinant chimeric enzymes, and constructed mutants were cultivated with 100 μg/mL of ampicillin at 37 °C in lysogenic broth medium. Their yeast transformants were incubated with SC-U medium (0.67% yeast nitrogen base without amino acids, 0.19% yeast synthetic dropout medium (without uracil), 2% glucose and 2% agar) at 20 °C and shaken at 200 rpm. Extra 2% galactose and 1% raffinose were added during induction. Fatty acids were purchased from Sigma-Aldrich (Shanghai, China). Restriction enzymes and other DNA-modifying reagents were obtained from Takara, Tiangen, and Sangon (Shanghai, China).

### 4.3. Sequence Comparison and Topology Prediction

FADS15 and oRiFADS17 desaturase were compared with 16 other reported desaturases using the Clustal Omega web server, ver. 6. The 18 sequences were aligned according to their substrate preference for LA or AA (Appendix A). They contained a series of enzymes from various strains, including *P. aphanidermatum*, *P. sojae*, *P. ramorum*, *P. infestans*, *S. diclina*, *P. parasitica*, *Octopus bimaculoides*, *Caenorhabditis elegans*, *R. irregularis*, *P. pastoris*, *M. alpina* 1S-4, *M. alpina* ATCC32222, *M. grisea*, *F. moniliforme*, *F. graminearum*, *S. kluyveri*, *P. frutescens*, and *S. hispanica*. The results of multiple sequence alignment revealed that there were three typical histidine box (his-box) motifs, HECGH (Ma110-114) and HDCGH (Ri125-129), HSKHH (Ma146-150) and HRHHH (Ri161-165), and HQCHH (Ma342-346) and HQIHH (Ri329-333), that interacted with the catalytic pocket underlined in red (Figure 1).

The predicted topologies of FADS15 and oRiFADS17 were modeled with the online servers TMHMM, HMMTOP, and TMpred [26]. The highly homologous structures of FADS15 and oRiFADS17 allowed swapping of the domains, which were separated by the specific hydrophobic transmembranes and his-box motifs.

### 4.4. PCR Amplifications and Plasmid Constructions of Chimeric Enzymes and Mutants

All primers for constructing plasmids of the parent proteins, chimeras, and mutants in this study are listed in Appendix A. The FADS15 and oRiFADS17 genes were amplified with primers including restrictive endonuclease sites *Eco*RI and *Xho*I. A set of fusion genes were also amplified by overlap extension PCR [18], and their generated fragments were then subcloned into yeast vector pYES2/NT C. For mutagenesis, corresponding primers were designed to introduce nucleotides into the two parent proteins for substitution of specific amino acids using the Fast Site-Directed Mutagenesis kit (Tiangen, Shanghai, China). All of the positive clones were confirmed by DNA sequencing (Sunny Biotechnology Co Ltd., Shanghai, China).

### 4.5. Yeast Transformation and Heterologous Expression in S. cerevisiae

The lithium acetate transformation method [13] was used to transform the constructed plasmids including pYES2-FADS15, pYES2-oRiFADS17, pYES2-Chimera 1–12 and mutants into the *INVSc1* yeast strain. Based on the SC-U selective plates that exhibited uracil prototrophy, their transformants were selected and identified by colony PCR.

To acquire insight into the substrate specificity of the ω-3Des, their transformants were cultivated for 24 h at 20 °C in 20 mL of SC-U medium. Then, 2% galactose, 1% raffinose and fatty acid substrates (0.1 mM cis-LA, 0.1 mM cis-GLA, 0.1 mM cis-DGLA, or 0.1 mM cis-AA) were supplemented into the induction buffer for another 24 h. Yeast cells were recovered by centrifugation, washed twice with distilled water and freeze-dried for fatty acid and protein analysis, and the remainder was kept at −80 °C for storage.

### 4.6. Lipid Extraction and Fatty Acid Analysis

Cell biomass was harvested by filtration, then washed twice with distilled water and frozen in liquid nitrogen. The constructed *S. cerevisiae INVSc1* strain with vector pYES2/NT C (pYES2) was used as negative control, and yeast transformants expressing the plasmids pYES2-FADS15 and pYES2-oRiFADS17 were used as positive control. The equivalent weight of freeze-dried biomass (50 mg) of lipids from the pYES2, the two parent proteins, the 12 chimeras, and the 8 mutants were extracted for statistical analyses.

Chloroform and methanol were used for lipid extraction and the extracted fatty acids were transesterified for GC analysis as described previously [27]. The resulting fatty acid methyl esters were then analyzed by an Agilent GC-2010 system (Shimadzu Co., Kyoto, Japan) coupled with a flame ionization detector and a DB-wax column (30 m × 0.32 mm). Samples were quantified with the following temperature programme: 40 °C for 5 min, 40–120 °C at 20 °C/min, 120 °C for 3 min, 120–190 °C at 5 °C/min, 190–220 °C at 4 °C/min and 220 °C for 20 min.

### 4.7. Homology Modelling of 3-Dimensional Structures and Molecular Docking of FADS15 and oRiFADS17

The crystal structure of human desaturase hSCD1 (PDB code: 4ZYO) was used to construct homology models of the 3-dimensional structure of FADS15 and oRiFADS17 [28]. The homology model was built from UCSF Chimera [29] graphical interface of Modeller 9.17 [30], with non-water HETATM (Zn2+ and stearoyl-CoA) included to keep the ligand binding pocket opened. Homology models were then preprocessed by Prepwizard [31] of Schrodinger 2015-4 suite by default parameters, and two ligands, LA-CoA and AA-CoA, were prepared by Ligprep as well. To further elucidate the molecular mechanism of their substrate specificities, LA-CoA and AA-CoA were docked into FADS15 and oRiFADS17 by using Glide SP [32] in Schrodinger 2015-4 suite, with a 1.0 Å core of the common substructure between these two ligands and stearoyl-CoA in hSCD1. The final docking conformation was selected by SP score and good contacts. UCSF Chimera and PYMOL were used to illustrate the docking results (Figure 2B and Figure 6). For FADS15 and oRiFADS17, a kink and the narrow aperture of the substrate tunnel and the relevant amino acid residues of the mutants are shown in different colors. Two metal zinc-ions bound to the substrate-binding pockets are also depicted as grey or yellow sphere.

### 4.8. Statistical Analyses

An analysis of variance (ANOVA) with repeated measurements was used to analyze differences of fatty acid compositions of the control (pYES2), the two parent proteins (pYES2-FADS15 and pYES2-oRiFADS17), the hybrid enzymes (12 chimeras) and the 8 mutants, and their conversion rates towards LA and AA substrates. Dunnett’s multiple comparisons were used as the post hoc test. GraphPad 7.0 (GraphPad Software, La Jolla, CA, USA) was used to perform statistical analyses. Data are presented as the mean of three samples ± standard error of the mean.

## 5. Conclusions

In this study, two ω-3 fatty acid desaturases, FADS15 and oRiFADS17 were confirmed to show a preference for LA and AA substrates, respectively. For the first time, the critical region between the his-boxes I and II was elucidated to determine the substrate preference of ω3Des by a comparative domain-swapping approach. The site-directed mutagenesis and simulation models were then used for further examination of crucial amino acids. Our results indicated that the crucial amino acids W129T and T144W mutations located in the upper part of hydrocarbon chain were involved in substrate preference, while the V137T and V152T mutations were confirmed to interfere with substrate recognition. Based on molecular docking, we visualized that how the proposed structure of ω3Des interacted with fatty acid substrates for the first time and interpreted the roles of crucial amino acids in regulating substrate specificity and enzyme activity. Collectively, this study provides significant insight into the molecular mechanism of the substrate specificity of ω3Des in regulating the metabolic flux of PUFAs, and the knowledge of these structure-function relationships may be conducive to the efficient production of desired fatty acids.

## Figures and Tables

**Figure 1 ijms-20-01603-f001:**
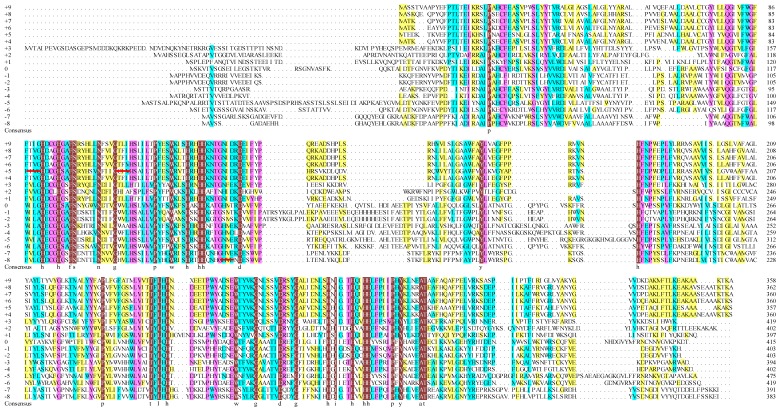
Multiple sequence alignment of ω-3 fatty acid desaturases. The 18 amino acid sequences were compared using the Clustal Omega Web server. They are a series of enzymes from various strains including *P. aphanidermatum*, *P. sojae*, *P. ramorum*, *P. infestans*, *S. diclina*, *P. parasitica*, *O. bimaculoides*, *C. elegans*, *R. irregularis*, *P. pastoris*, *M. alpina* 1S-4, *M. alpina* ATCC32222, *M. grisea*, *F. moniliforme*, *F. graminearum*, *S. kluyveri*, *P. frutescens*, and *S. hispanica* (Appendix A). For each sequence, a GenBank accession no. is provided. The sequences underlined in red are three conserved his-box motifs (one HX_3_H and two HX_2_HH).

**Figure 2 ijms-20-01603-f002:**
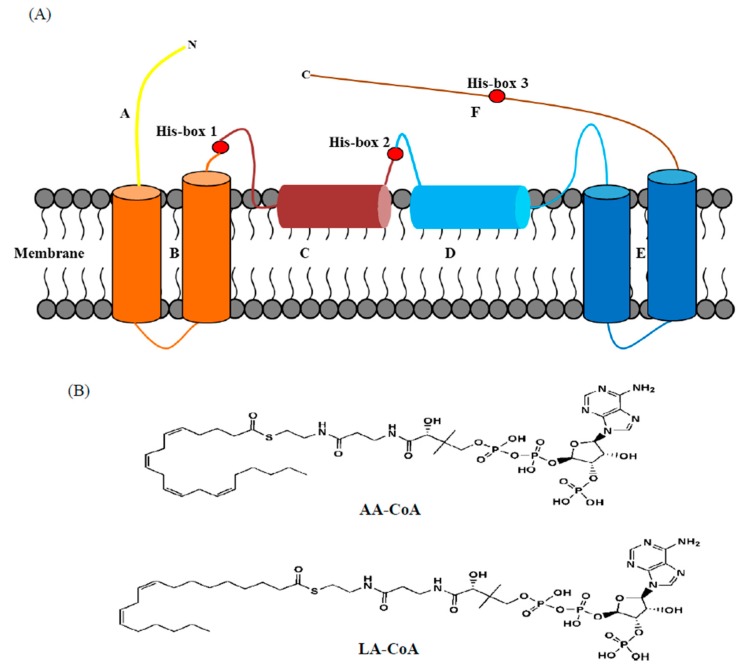
Proposed topology model for FADS15 and oRiFADS17 desaturase (**A**). TM 1–4 indicate four transmembrane domains and AH 1–2 indicate spanning helices entrapped in the membrane. The red pellets denote three his-box motifs and the letters A–F designate the regions used for constructing fused enzymes labeled with various colors. The putative model was predicted by online servers TMHMM, HMMTOP, and TMpred. The plane graph (**B**) indicates LA-CoA and AA-CoA substrates used for molecular docking with homology models of FADS15 and oRiFADS17.

**Figure 3 ijms-20-01603-f003:**
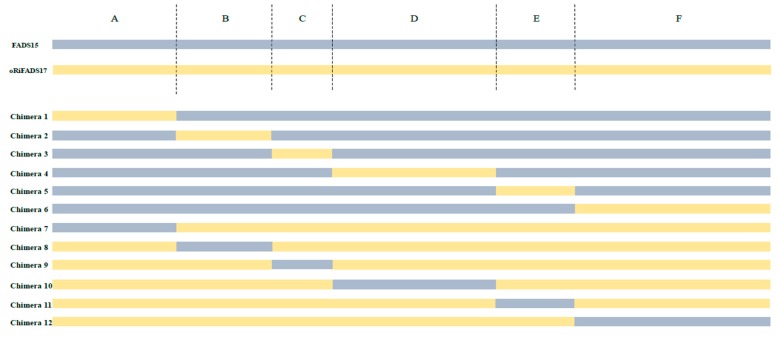
Schematic structural diagrams of the fused chimeras 1–12. The blue rectangle indicates the amino acid sequence of FADS15 desaturase and the orange rectangle indicates that of oRiFADS17 desaturase. Their overall amino acid sequences are represented above the rectangles of 12 chimeric enzymes, which are divided into six parts (A–F) by the specific hydrophobic domains and his-boxes. This work was created using Microsoft Word, Clustal Omega, and Adobe Illustrator.

**Figure 4 ijms-20-01603-f004:**
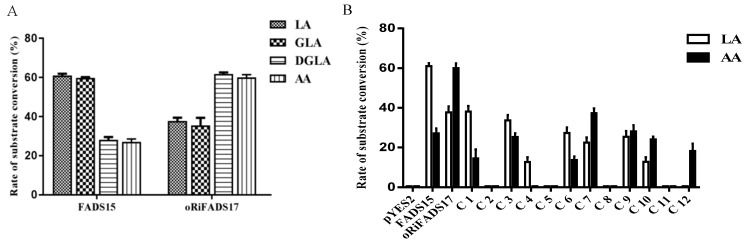
Verification of the conversion rates (**A**) of yeast transformants harboring plasmids pYES2-FADS15 and pYES2-oRiFADS17 towards various ω-6 fatty acids substrates, including LA, GLA, DGLA, and AA. Relative conversion rates (**B**) of each recombinant protein FADS15, oRiFADS17, and Chimeras 1–12. The yeast transformant with pYES2/NT C vector was used as the control strain. Substrate conversion rate = 100 × [product/(product + substrate)].

**Figure 5 ijms-20-01603-f005:**
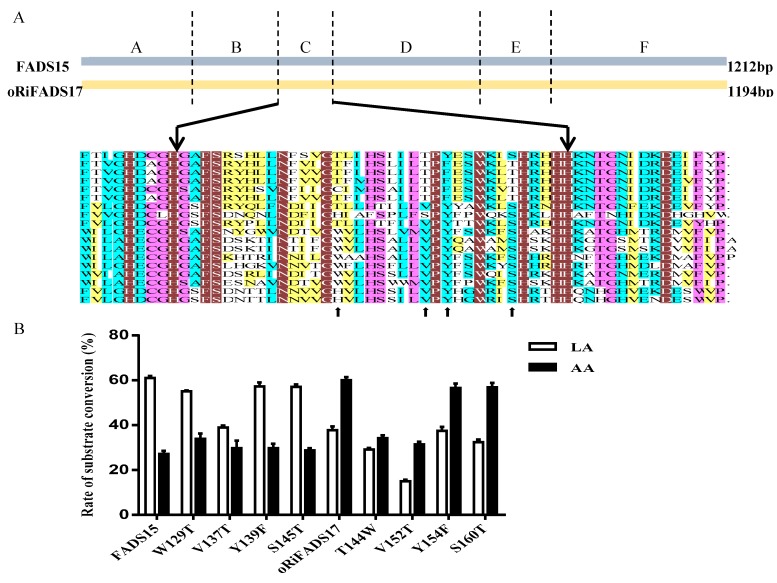
A partial comparison of amino acid sequences of FADS15 and oRiFADS17 (**A**) between the his-boxes I and II from 18 species listed in Appendix A. Black lines indicate the boundaries of the 36 amino acids in domain C. Arrows below the alignment indicate the positions of amino acids that were selected for site-directed mutagenesis; relative substrate conversion rates of each mutant (**B**). 0.1mM LA and 0.1 mM AA were added in yeast cultures after induction with 2% galactose. Substrate conversion rate = 100 × [product/(product + substrate)].

**Figure 6 ijms-20-01603-f006:**
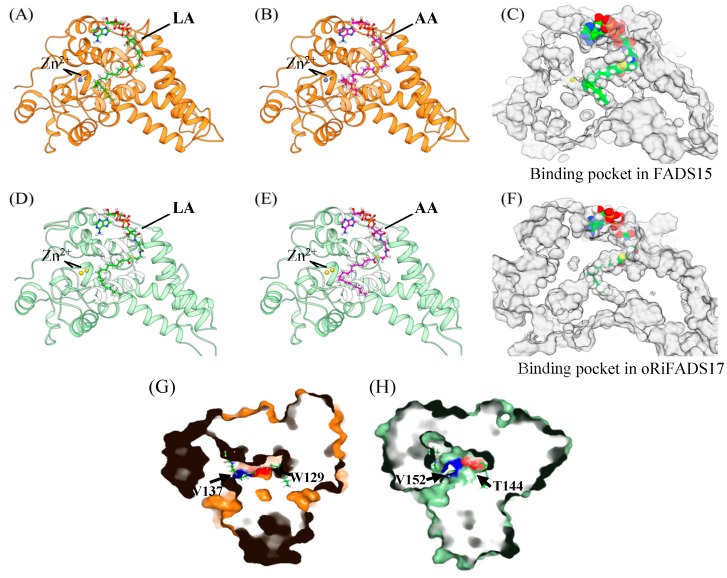
Homology modeling of 3-dimensional structures and molecular docking of FADS15 and oRiFADS17. On the basis of the crystal structure information of human stearoyl-CoA desaturase hSCD1, the LA-CoA (**A**), AA-CoA (**B**) substrates and the binding pocket (**C**) were clearly shown in the simulation model of FADS15 using the software PYMOL; The LA-CoA (**D**), AA-CoA (**E**) substrates and the binding pocket (**F**) were also displayed in the simulation model of oRiFADS17. The crucial amino acid residues V137 and W129 of FADS15 (**G**), V152 and T144 of oRiFADS17 (**H**) were indicated by arrows and labeled in blue or red in the cross-sections of electrostatic surface of the two desaturases.

**Table 1 ijms-20-01603-t001:** Fatty acid compositions (% *w*/*w*) and conversion rates (%) of yeast transformants expressing the control, the wild-type pYES2-FADS15 and pYES2-oRiFADS17 ^α^.

	Transformants	pYES2 (Control)		pYES2-FADS15		pYES2-oRiFADS17	
Fatty acid		+ LA	+ AA	+ LA	+ AA	+ LA	+ AA
16:0 (PA)		25.4 ± 1.4 ^a^	25.6 ± 1.1 ^a^	23.3 ± 1.3 ^a^	24.3 ± 1.5 ^a^	25.8 ± 2.0 ^a^	22.9 ± 2.6 ^a^
16:1 (PA)		33.4 ± 1.7 ^a^	33.9 ± 3.1 ^a^	32.2 ± 1.8 ^a^	34.0 ± 1.8 ^a^	30.1 ± 1.6 ^a^	32.2 ± 3.2 ^a^
18:0 (SA)		4.5 ± 0.4 ^a^	4.7 ± 0.5 ^a^	4.6 ± 0.4 ^a^	4.3 ± 0.4 ^a^	5.1 ± 0.3 ^a^	5.0 ± 0.6 ^a^
18:1 (OA)		26.0 ± 2.1 ^a^	25.1 ± 1.1 ^a^	26.6 ± 1.1 ^a^	27.2 ± 2.6 ^a^	26.7 ± 1.5 ^a^	28.6 ± 3.5 ^a^
18:2 (LA, ω-6)		8.6 ± 0.5 ^a^	ND	3.2 ± 0.1 ^c^	ND	4.7 ± 0.3 ^b^	ND
18:3 (GLA, ω-3)		ND	ND	4.9 ± 0.3 ^a^	ND	2.9 ± 0.3 ^b^	ND
20:4 (AA, ω-6)		ND	5.8 ± 0.2 ^a^	ND	4.3 ± 0.3 ^b^	ND	2.0 ± 0.1 ^c^
20:5 (EPA, ω-3)		ND	ND	ND	1.6 ± 0.1 ^b^	ND	3.0 ± 0.3 ^a^
LA conversion rate		0.0 ± 0.0 ^c^	-	61.0 ± 1.6 ^a^	-	37.8 ± 2.9 ^b^	-
AA conversion rate		-	0.0 ± 0.0 ^c^	-	27.2 ± 2.6 ^b^	-	60.1 ± 2.4 ^a^

^α^ Values are mean of three samples ± standard error of the mean. Values with the same letter in the same row are not significantly different. a, *p*-value < 0.05, b, *p*-value < 0.01, c, *p*-value< 0.001. Conversion rate = 100 × [product/(product + substrate)]. ND: not detected. Substrate concentration was 0.1 mM linoleic acid (LA, C18:2^Δ9,12^) or arachidonic acid (AA, C20:4^Δ5,8,11,14^).

**Table 2 ijms-20-01603-t002:** Fatty acid compositions (% *w*/*w*) and conversion rates (%) of yeast transformants expressing chimeric enzymes ^α^.

Fatty Acid	FADS15 Chimeras						oRiFADS17 Chimeras					
	C1	C2	C3	C4	C5	C6	C7	C8	C9	C10	C11	C12
16:0 (PA)	25.3 ± 1.4 ^a^	21.8 ± 2.1 ^a^	24.4 ± 1.0 ^a^	24.6 ± 1.6 ^a^	23.6 ± 1.2 ^a^	20.2 ± 1.1 ^b^	23.7 ± 2.2 ^a^	23.0 ± 2.4 ^a^	24.7 ± 1.1 ^a^	22.1 ± 2.6 ^a^	22.1 ± 0.8 ^a^	22.1 ± 2.7 ^a^
16:1 (PA)	30.4 ± 2.2 ^a^	31.6 ± 1.5 ^a^	30.0 ± 1.7 ^a^	31.2 ± 1.2 ^a^	32.4 ± 2.4 ^a^	32.1 ± 1.9 ^a^	30.4 ± 0.8 ^a^	31.4 ± 1.3 ^a^	30.8 ± 3.4 ^a^	33.4 ± 2.4 ^a^	30.7 ± 3.2 ^a^	29.7 ± 1.4 ^a^
18:0 (SA)	4.0 ± 0.4 ^b^	4.8 ± 0.3 ^a^	4.7 ± 0.4 ^a,b^	4.0 ± 0.5 ^b^	5.0 ± 0.4 ^a,b^	4.9 ± 0.3 ^a,b^	5.3 ± 0.3 ^a,b^	5.3 ± 1.0 ^a,b^	4.8 ± 0.6 ^a,b^	4.6 ± 0.4 ^b^	5.7 ± 0.3 ^a^	5.2 ± 0.3 ^a,b^
18:1 (OA)	23.1 ± 1.7 ^a^	23.6 ± 1.4 ^a^	22.3 ± 1.6 ^a,b^	22.2 ± 1.1 ^a,b^	21.6 ± 0.9 ^a,b^	24.4 ± 1.8 ^a^	21.6 ± 2.3 ^a,b^	20.9 ± 1.8 ^a,b^	18.6 ± 2.2 ^b^	23.4 ± 0.9 ^a^	21.3 ± 2.3 ^a,b^	22.8 ± 1.7 ^a^
18:2 (LA, ω-6)	4.6 ± 0.3 ^c^	8.4 ± 1.1 ^a^	5.2 ± 0.3 ^c^	7.2 ± 0.3 ^b^	7.7 ± 0.5 ^a,b^	5.6 ± 0.5 ^c^	5.4 ± 0.5 ^c^	7.9 ± 0.4 ^a,b^	6.5 ± 0.1 ^b,c^	6.9 ± 0.1 ^b^	8.5 ± 0.4 ^a^	8.4 ± 0.4 ^a^
18:3 (GLA, ω-3)	2.8 ± 0.2 ^a^	ND	2.7 ± 0.2 ^a,b^	1.0 ± 0.2 ^d^	ND	2.1 ± 0.1 ^b^	1.6 ± 0.1 ^c^	ND	2.2 ± 0.4 ^b^	1.0 ± 0.2 ^d^	ND	ND
20:4 (AA, ω-6)	4.1 ± 0.3 ^b^	5.6 ± 0.8 ^a,b^	4.2 ± 0.2 ^b^	5.0 ± 0.5 ^b^	5.4 ± 0.3 ^a,b^	4.7 ± 0.3 ^b^	4.3 ± 0.6 ^b^	5.7 ± 0.4 ^a,b^	4.4 ± 0.4 ^b^	4.1 ± 0.2 ^b^	6.3 ± 0.4 ^a^	5.1 ± 0.8 ^b^
20:5 (EPA, ω-3)	0.7 ± 0.2 ^d^	ND	1.4 ± 0.1 ^b,c^	ND	ND	0.8 ± 0.1 ^d^	2.6 ± 0.2 ^a^	ND	1.7 ± 0.3 ^b^	1.3 ± 0.1 ^c^	ND	1.2 ± 0.1 ^c^
LA conversion rate	38.1 ± 2.8 ^a^	-	33.7 ± 2.7 ^a^	12.7 ± 2.3 ^c^	-	27.4 ± 2.8 ^b^	22.5 ± 2.7 ^b^	-	25.5 ± 2.9 ^b^	12.9 ± 2.3 ^c^	-	-
AA conversion rate	14.5 ± 4.5 ^c^	-	25.4 ± 1.9 ^b^	-	-	13.7 ± 1.9 ^c^	37.4 ± 2.3 ^a^	-	28.1 ± 3.1 ^b^	24.1 ± 1.5 ^b,c^	-	18.2 ± 3.7 ^c^

^α^ Values are mean of three samples ± standard error of the mean. Values with the same letter in the same row are not significantly different. a, *p*-value < 0.05, b, *p*-value < 0.01, c, *p*-value< 0.001. Conversion rate = 100 × [product/(product + substrate)]. ND: not detected. Substrate concentration was 0.1 mM LA or AA.

**Table 3 ijms-20-01603-t003:** Fatty acid compositions (% *w*/*w*) and conversion rates (%) of yeast transformants expressing the mutant desaturases ^α^.

Fatty Acid	FADS15 Mutants				oRiFADS17 Mutants			
	W129T	V137T	Y139F	S145T	T144W	V152T	Y154F	S160T
16:0 (PA)	23.9 ± 0.7 ^a,b^	23.2 ± 0.7 ^a,b^	23.4 ± 2.4 ^a,b^	25.3 ± 1.1 ^a^	23.9 ± 1.5 ^a,b^	20.9 ± 2.0 ^b^	22.7 ± 1.8 ^a,b^	23.0 ± 1.0 ^a,b^
16:1 (PA)	30.8 ± 2.2 ^a^	34.1 ± 2.4 ^a^	30.7 ± 3.1^a^	31.2 ± 2.8^a^	32.1 ± 2.1 ^a^	31.8 ± 2.3 ^a^	30.1 ± 3.1 ^a^	31.1 ± 3.9 ^a^
18:0 (SA)	5.6 ± 0.6 ^a^	3.7 ± 0.2 ^b^	4.6 ± 0.9 ^a,b^	4.6 ± 0.8 ^a,b^	5.3 ± 0.5 ^a^	5.3 ± 0.4 ^a^	4.9 ± 0.4 ^a,b^	4.8 ± 0.5 ^a,b^
18:1 (OA)	21.1 ± 2.0 ^a^	23.3 ± 2.2 ^a^	22.1 ± 2.0 ^a^	21.8 ± 1.4 ^a^	20.9 ± 1.7 ^a^	23.8 ± 2.2 ^a^	23.2 ± 2.4 ^a^	22.2 ± 2.1 ^a^
18:2 (LA, ω-6)	3.7 ± 0.3 ^c^	4.3 ± 0.1 ^b,c^	3.6 ± 0.3 ^c^	3.5 ± 0.1 ^c^	5.6 ± 0.2 ^a^	6.1 ± 0.4 ^a^	4.9 ± 0.3 ^b^	5.6 ± 0.4 ^a^
18:3 (GLA, ω-3)	4.6 ± 0.3 ^a^	3.2 ± 0.2 ^b^	4.9 ± 0.2 ^a^	4.6 ± 0.5 ^a^	2.3 ± 0.2 ^c^	1.1 ± 0.1 ^d^	2.9 ± 0.2 ^b^	2.7 ± 0.1 ^b^
20:4 (AA, ω-6)	3.8 ± 0.2 ^a^	3.6 ± 0.2 ^a^	3.8 ± 0.3 ^a^	3.6 ± 0.4 ^a^	3.4 ± 0.3 ^a^	3.9 ± 0.5 ^a^	2.6 ± 0.1 ^b^	2.5 ± 0.3^b^
20:5 (EPA, ω-3)	2.0 ± 0.4 ^b^	1.3 ± 0.2 ^b^	1.6 ± 0.3 ^b^	1.5 ± 0.1 ^b^	1.8 ± 0.3 ^b^	1.8 ± 0.2 ^b^	3.4 ± 0.4 ^a^	3.3 ± 0.2 ^a^
LA conversion rate	55.1 ± 0.5 ^a^	42.4 ± 2.0 ^b^	57.2 ± 3.0 ^a^	57.0 ± 2.0 ^a^	29.2 ± 1.1 ^c^	15.0 ± 1.2 ^d^	37.4 ± 3.2 ^b^	32.5 ± 1.9 ^c^
AA conversion rate	33.9 ± 4.1 ^b^	28.0 ± 1.2 ^b^	29.7 ± 3.5 ^b^	28.7 ± 1.8 ^b^	34.2 ± 2.1 ^b^	31.3 ± 2.2 ^b^	56.5 ± 3.4 ^a^	56.8 ± 3.4 ^a^

^α^ Values are mean of three samples ± standard error of the mean. Values with the same letter in the same row are not significantly. a, *p*-value < 0.05, b, *p*-value < 0.01, c, *p*-value< 0.001. Conversion rate = 100 × [product/(product + substrate)]. Substrate concentration was 0.1 mM LA or AA.

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
