# Peer review of "Structural Determinants of Substrate Specificity of Omega-3 Desaturases from Mortierella alpina and Rhizophagus irregularis by Domain-Swapping and Molecular Docking"

_ijms, 2019, doi:10.3390/ijms20071603_

Round 1
Reviewer 1 Report
Dear Authors,
After the review process, I have several comments:
- the authors should insert statistical significance in tables; the authors should insert a statistical section at the end of section 4;
- the authors should clearly present/describe the control in section 4;
- the authors should detail the limitative factors of the study in section 3;
- the authors should expand the practical applications of these researches, page 1, first paragraph - section 1;
- the authors should detail in section 3, the possible valorisation of the relationship between the structure and function of ω3Des.
Best regards!
Author Response
Major comments
Comment(1): the authors should insert statistical significance in tables; the authors should insert a statistical section at the end of section 4.
Response: Thank you for the suggestion. We have inserted required statistical significance in all tables throughout the text and a statistical section at the end of section 4 as follows: "Analysis of variance (ANOVA) with repeated measurements was used to analyse differences of fatty acid compositions of the control (pYES2), the two parent proteins (pYES2-FADS15 and pYES2-oRiFADS17), the hybrid enzymes (12 chimeras) and the 8 mutants, and their conversion rates towards LA and AA substrates. Dunnett's multiple comparisons were used as the post hoc test. Graphpad 7.0 was used to perform the statistical analyses. Data are presented as mean of three samples ± standard error of the mean." (Line 4-10, page 12)
Comment(2): the authors should clearly present/describe the control in section 4.
Response: Thank you for your reminder. To be more accurate and clear, we have concisely described the control according to experimental procedures in the corresponding paragraph of section 4 as follows: "Cell biomass was harvested by filtration, then washed twice with distilled water and frozen in liquid nitrogen. The constructed S. cerevisiae INVSc1 strain with vector pYES2/NT C (pYES2) was used as negative control, and yeast transformants expressing the plasmids pYES2-FADS15 and pYES2-oRiFADS17 were used as positive controls. The equivalent weight of freeze-dried biomass (50 mg) of lipids from the pYES2, the two parent proteins, the 12 chimeras and the 8 mutants were extracted for statistical analyses." (Line 24-29, page 11)
Comment(3): the authors should detail the limitative factors of the study in section 3.
Response: Thank you for your valuable suggestion. We have detailed the limitative factors of the study in section 3 as follows "Although various n-3 fatty acid desaturases have been identified, few Δ17Des have been isolated compared to reported Δ15Des. The reported Δ17Des such as sdd17 [13], Pi-D17 [14], and oPPD17 [15] show moderate similarities in their primary sequences with FADS15 (Δ15Des). However, they still could not be applied to conduct domain-swapping with FADS15 because of their differences in lengths of amino acid sequences. Despite considerable efforts to identify and characterise ω3Des from various species (Supplemental Table S2), the underlying cause of the distinct substrate specificity of ω3Des has not been elucidated due to difficulties in separation and purification of large quantities of membrane proteins and a lack of 3-dimensional crystal structure information of ω3Des." (Line 1-9, page 9).
Comment(4): the authors should expand the practical applications of these researches, page 1, first paragraph - section 1.
Response: Thank you for the suggestion. We have expanded the practical applications of these researches as follows "They have been used to produce eicosapentaenoic acid (EPA, C20:5Δ5,8,11,14,17) and docosahexaenoic acid (DHA, C22:6Δ4,7,10,13,16,19) in oleaginous microorganisms, which are important for brain development, tissue formation and repair in mammals and also have significant effects in the prevention of asthma, cancer, depression, obesity and immune cardiovascular disorder [2-4]." (Line 40-44, page 1)
Comment(5): the authors should detail in section 3, the possible valorisation of the relationship between the structure and function of ω3Des.
Response: Thank you for the suggestion. We have added the description of the possible valorisation of this study at the end of section 3 as follows "Traditional source of n-3 PUFAs is diminishing due to overfishing and marine pollution, and a low dietary intake of n-3 PUFAs, especially EPA and DHA, is observed worldwide. The knowledge of relationship between the structure and function of ω3Des could be conducive for molecular modification of desaturases and genetic engineering of oleaginous microorganisms, and therefore contribute to produce PUFAs for dietary supplementation or the prevention and treatment of numerous diseases." (Line 15-20, page 10)
Reviewer 2 Report
This manuscript describes the study of omega-3 fatty acid desaturases to understand the substrate specificity. The authors chose FADS15 and oRiFADS17 as target proteins and used domain swapping and site directed mutagenesis. In order to look at the structural significance, the authors employed homology modeling and docking between enzymes and substrates. Overall, the research was done thoroughly, and the manuscript was concisely written. The research results, however, are rather convoluted. Some suggestions are as follows:
1. A typical, graphical illustration of the rate should be shown, (e.g., time vs [product] or [substrate] vs rate).
2. The details of the enzyme assay information should be included in the method section. Specifically, the authors mentioned ‘high fatty acid catalytic efficiency’. How did the authors deduce the catalytic efficiency? In the text, there are rates but no description of the catalytic efficiency information.
3. The method section fails to describe the details of computational docking. Perhaps the reader should infer that the authors used Chimera software to conduct the docking, but no confirming details are given.
4. The conclusion section should include key items discovered from this research. This information would clarify and summarize this manuscript,(e.g., site-directed mutagenesis results should be mentioned).
Author Response
Comment: This manuscript describes the study of omega-3 fatty acid desaturases to understand the substrate specificity. The authors chose FADS15 and oRiFADS17 as target proteins and used domain swapping and site directed mutagenesis.……
Response: Many thanks to your suggestions and your recognition of our work. We have studied them carefully and tried our best to revise the manuscript.
Major comments
Comment(1): A typical, graphical illustration of the rate should be shown, (e.g., time vs [product] or [substrate] vs rate).
Response: Thank you for the suggestion. We completely agree with this valuable suggestion by the reviewer. As a matter of fact, we attempted to carry out such an analysis. According to our analysis and the literature, the enzyme assay system is much more complicated than other cytosolic enzymes. Furthermore, we have not finished the successful large-scale overexpression, separation and purification of active membrane desaturases in vitro. In our future study, we hope to finish enzymatic activity and Kinetic analysis of ω3Des and their mutants.
Comment(2): The details of the enzyme assay information should be included in the method section. Specifically, the authors mentioned ‘high fatty acid catalytic efficiency’. How did the authors deduce the catalytic efficiency? In the text, there are rates but no description of the catalytic efficiency information.
Response: Thank you for the suggestion. We actually did not do the enzyme assay of ω3Des in vitro in this research, and the description "high fatty acid catalytic efficiency" mentioned in section 3 was conjectural based on their high conversion rates towards the same concentration of corresponding substrates and the same culture conditions. We are sorry that this part was not clear in the original manuscript. We should have explained that. To avoid misunderstanding, we have revised the contents of catalytic efficiency to conversion rates throughout the text.
"but their conversion efficiencies were much lower" to "but their conversion rates were much lower" (Line 10, page 5).
"the conversion efficiency of W129T" to "the conversion rates of W129T" (Line 6, page 7).
"possess high fatty acid catalytic efficiency" to "possess high fatty acid conversion rates" (Line 12, page 9).
Comment(3): The method section fails to describe the details of computational docking. Perhaps the reader should infer that the authors used Chimera software to conduct the docking, but no confirming details are given.
Response: Thank you for your valuable suggestion. We have overwritten and detailed the steps of computational docking in section 4, and added relevant references as follows "The crystal structure of human desaturase hSCD1 (PDB code: 4ZYO) was used to construct homology models of the 3-dimensional structure of FADS15 and oRiFADS17 [25]. Homology model was built from UCSF Chimera [31] graphical interface of Modeller 9.17 [32], with non-water HETATM (Zn2+ and stearoyl-CoA) included to keep ligand binding pocket opened. Homology models were then preprocessed by Prepwizard [33] of Schrodinger 2015-4 sutie by default parameters, and two ligands, LA-CoA and AA-CoA, were prepared by Ligprep as well. To further elucidate the molecular mechanism of their substrate specificities, LA-CoA and AA-CoA was docked into FADS15 and oRiFADS17 by using Glide SP [34] in Schrodinger 2015-4 suite, with a 1.0 Å core of the common substructure between these two ligands and stearoyl-CoA in hSCD1. The final docking conformation was selected by SP score and good contacts. UCSF Chimera and PYMOL were used to illustrate the docking results (Figures 2B and 6). For FADS15 and oRiFADS17, a kink and the narrow aperture of the substrate tunnel and the relevant amino acid residues of the mutants are shown in different colors. Two metal zinc-ions bound to the substrate-binding pockets are also depicted as grey or yellow sphere." (Line 37-47, page 11).
Comment(4): The conclusion section should include key items discovered from this research. This information would clarify and summarize this manuscript,(e.g., site-directed mutagenesis results should be mentioned).
Response: Thank you for the suggestion. We have added the description of the site-directed mutagenesis results as follows "The site-directed mutagenesis and simulation models were then used for further examination of crucial amino acids. Our results indicated that the crucial amino acids W129T and T144W mutations located in the upper part of hydrocarbon chain were involved in substrate preference, while the V137T and V152T mutations were confirmed to interfere with substrate recognition.” (Line 15-19, page 12).
Round 2
Reviewer 1 Report
Dear Author,
I do not have any supplementary comments.
Best regards!
Reviewer 2 Report
Much improved. It looks fine for publication.